# New Therapeutic Maneuver for Horizontal Semicircular Canal Cupulolithiasis: A Prospective Randomized Trial

**DOI:** 10.3390/jcm11144136

**Published:** 2022-07-16

**Authors:** Dong-Han Lee, Joon Yong Park, Tae Hee Kim, Jung Eun Shin, Chang-Hee Kim

**Affiliations:** Department of Otorhinolaryngology-Head and Neck Surgery, Konkuk University Medical Center, Research Institute of Medical Science, Konkuk University School of Medicine, Seoul 05030, Korea; 20200189@kuh.ac.kr (D.-H.L.); 20180105@kuh.ac.kr (J.Y.P.); 20200135@kuh.ac.kr (T.H.K.); 20050055@kuh.ac.kr (J.E.S.)

**Keywords:** horizontal semicircular canal, benign paroxysmal positional vertigo, apogeotropic, cupulolithiasis, treatment outcome

## Abstract

Background: There are debates on whether mastoid oscillation has any benefit or harm in treating horizontal semicircular canal (HSCC) cupulolithiasis. The goal of this study was to investigate the therapeutic effects of the new maneuver using only inertia and gravity and compare it with the previously reported cupulolith repositioning maneuver using mastoid vibration (CuRM). Methods: We enrolled 57 patients diagnosed with HSCC cupulolithiasis. Patients were randomly allocated to the previously reported CuRM or the new maneuver (briefly, 30° head rotation to the affected side and thereafter bidirectional side-lying) using simply inertia and gravity, and their immediate and short-term effects were evaluated. Results: The immediate success rate did not differ significantly between the CuRM (8 of 22, 36.4%) and the new maneuver (10 of 35, 28.6%) groups (*p* = 0.538, Pearson’s chi-square test). The late resolution rates at the first follow-up of the CuRM (75%, 9 of 12) and new maneuver groups (82.6%, 19 of 23) were very high, and there was no statistical difference between them (*p* = 0.670, Fisher’s exact test). Conclusions: This study showed that the new maneuver was effective for treating HSCC cupulolithiasis with an immediate success rate of 28.6% (10 of 35). Although it did not show better results than the existing maneuver using vibration, there was no statistical difference. Considering the debate on the effectiveness of oscillation, we believe our new maneuver is a conservative alternative that uses only inertia and gravity, and it can be easily performed in clinics where oscillation equipment is not available.

## 1. Introduction

Benign paroxysmal positional vertigo (BPPV) is the leading cause of vestibular disorders [1]. According to recent demographic studies, posterior canal-BPPV (PC-BPPV) was most common (45–49%), followed by horizontal semicircular canal (HSCC) canalolithiasis (15–22.8%) and HSCC cupulolithiasis (14.5–40%) [2,3]. Although several treatment maneuvers for PC-BPPV (e.g., modified Epley maneuver or Semont maneuver) or HSCC canalolithiasis (e.g., Barbeque roll maneuver, Gufoni maneuver, and Asprella maneuver…) are known to have a high treatment success rate (50–over 90%) [1], the immediate treatment of HSCC cupulolithiasis is often challenging (32%, according to a recent study) [3]. HSCC cupulolithiasis is characterized by positional vertigo and persistent direction-changing positional nystagmus (DCPN) beating to the uppermost (apogeotropic) ear induced by turning the head to the right or left in a supine position (supine head roll test), while in HSCC canalolithiasis of the posterior arm, DCPN beats to the lowermost (geotropic) ear during the supine head roll test [4]. Apogeotropic HSCC BPPV is believed to be caused by either otolith debris attached to the cupula (cupulolithiasis) or free-floating debris within the anterior arm of the HSCC (canalolithiasis) [4]. Generally, HSCC cupulolithiasis and canalolithiasis of the anterior arm of HSCC may be differentiated by their nystagmus characteristics, such as duration, latency, and fatigability during the supine head roll test. Apogeotropic DCPN in cupulolithiasis is persistent and lacks latency or fatigability, whereas, in canalolithiasis, nystagmus is transient and shows latency or fatigue [5,6,7,8]. Considering these two variants of apogeotropic HSCC BPPV, clinicians can choose a treatment maneuver that is effective for both variants or effective for an individual variant.

In HSCC cupulolithiasis, it is also presumed that otolith debris can be attached to either the utricular or the canal side of the cupula. However, clinicians cannot differentiate between them during the diagnostic head roll test. Therefore, the ideal treatment maneuver should be designed considering the possibility of both cases. Until now, several treatment maneuvers have been proposed to treat the HSCC cupulolithiasis regardless of the location where the otolithic debris is attached to the cupula, including a head shaking maneuver in a supine position, a cupulolith repositioning maneuver by mastoid oscillation (CuRM) [9], head-tilt hopping exercises [10], and, recently, Zuma’s maneuver [11]. However, the effectiveness of each treatment method has not been sufficiently verified in a large number of patients, and no standard maneuver has been established. In general, the repositioning maneuver for HSCC cupulolithiasis relies on mechanisms such as mastoid oscillation, gravity, and inertia (rapid acceleration or deceleration) to detach the otolith debris from the cupula and move it into the utricle. However, there may be a controversy that mastoid vibration can rather induce the unintentional detachment of the intact otoconia from the utricular macula [12,13,14,15], and hopping exercises may be difficult for elderly patients with limited physical activity or patients with severe symptoms.

Here, we introduce a new treatment maneuver for treating HSCC cupulolithiasis without using a vibrator or hopping exercise, based on a mechanism using only inertia and gravity, considering both the canal-side and the utricular-side debris. The aim of this study was to investigate the therapeutic effects of the new maneuver, and we compared the effects with the previously introduced maneuver using mastoid oscillation.

## 2. Patients and Methods

### 2.1. Patients

We enrolled 57 patients with HSCC cupulolithiasis who visited the emergency room or outpatient clinic of Konkuk University Medical Center, Seoul, Korea, between September 2020 and January 2022. The criteria for inclusion in this study were: (1) complaining of positional vertigo and (2) persistent direction-changing horizontal nystagmus toward the uppermost ear (apogeotropic nystagmus) in the supine head roll test. The exclusion criteria were: (1) otologic symptoms suggesting middle ear or other labyrinthine disorders; (2) recent history of labyrinthine disorders such as acute unilateral vestibulopathy, sudden sensorineural hearing loss, Meniere’s disease, or central nervous system disorders; and (3) BPPV patients with multiple canal involvement or those with a transition to HSCC cupulolithiasis from other subtypes of BPPV during or after a positional nystagmus test or treatment maneuver.

All enrolled patients were evaluated for spontaneous and gaze-evoked nystagmus, saccades and smooth pursuit, limb ataxia, and balance function. There was no identifiable neurologic deficit in the participants in the initial evaluation and during the follow-up. The study design was explained to the patients, and written informed consent was obtained from each patient. The study protocol was approved by the Institutional Review Board (KUMC 2020-07-079-002).

### 2.2. Diagnosis of HSCC Cupulolithiasis and Lateralization

All patients underwent positioning maneuvers including bowing, lying down, supine head roll, Dix-Hallpike, and straight head hanging position. During the positioning maneuvers, nystagmus was observed without eyeball fixation using a video-Frenzel goggle device (SLMED, Seoul, Korea) [16]. A typical HSCC cupulolithiasis was diagnosed by the following nystagmus findings: (1) direction-changing horizontal nystagmus beating towards the uppermost ear (apogeotropic nystagmus) in both right and left head turning in the supine position, (2) persistent apogeotropic nystagmus without latency in the supine head roll test to exclude canalolithiasis of the anterior arm of HSCC, and (3) no evidence of prominent vertical or torsional nystagmus component suggesting BPPV involving the anterior or posterior semicircular canal.

The affected side was determined with the following assumptions: (1) the intensity of the apogeotropic nystagmus is weaker when the head is turned to the affected side according to Ewald’s second law [8,17,18]; (2) the lying-down nystagmus beats mostly towards the affected side, while the bowing nystagmus beats mostly towards the healthy side [8,17,19]; and (3) a null point, where the nystagmus is suppressed, is mostly observed on the affected side with a 15- to 30-degree head turn [16].

### 2.3. Treatment Procedures and Treatment Effects Evaluation

This study used a prospective randomized trial. We prospectively investigated the immediate and short-term treatment effects of the CuRM and the new maneuver. Patients were randomized into either the CuRM or new maneuver group by a computerized random number function in Microsoft Excel 2019 (Microsoft Crop.; Redmond, WA, USA). 

The CuRM was performed as follows [9]: (1) at the beginning, the patient was placed in the supine position; (2) the head was turned 135° to the affected side (1st position, the body was moved from supine to lateral decubitus to the affected side), then oscillation was applied at the suprameatal triangle in the posterior superior area of the affected side auricle with a 60 Hz hand-held vibrator for 30 s to detach the otoliths from the cupula (canal-side debris may be detached); (3) the head was turned 45° to the healthy side (2nd position, lateral decubitus to the affected side); (4) the head was turned 90° to the healthy side (3rd position, supine position); (5) the head was turned 90° to the healthy side (4th position, lateral decubitus to the healthy side), then oscillation was applied if apogeotropic nystagmus was observed (utricular-side debris may be detached); (6) the head was turned 90° in the same direction (5th position, prone position), then the patient was slowly returned to a sitting position.

The new maneuver was performed as follows: (1) the patient was placed in the sitting position in the center of the examination table (Figure 1A); (2) the head was turned 30° to the affected side to align the affected side cupula axis to the sagittal plane (1st position, Figure 1B); (3) the patient was quickly brought down on the healthy side and held for two minutes in that position (2nd position, Figure 1C). Initial brisk healthy side downward acceleration of the affected side cupula may detach the canal-side debris from the cupula, and final brisk deceleration of the cupula may detach the utricular-side debris from the cupula. Two minutes of waiting at this step may help the utricular-side debris fall into the utricle by gravity. (4) The patient was slowly returned to the sitting position and kept in that position for 1 min (3rd position, Figure 1D); (5) the head was turned 30° to the affected side, then the patient was rapidly brought down on the affected side and held for two minutes (4th position, Figure 1E). Initial brisk affected side downward acceleration of the cupula may detach the utricular-side debris from the cupula, and final brisk deceleration of the cupula may detach the canal-side debris from the cupula. Two minutes of waiting at this step may help the canal-side debris fall into the canal by gravity. (6) The head was quickly turned 45° upward and kept in that position for 2 min to displace otolith debris of the anterior arm into the posterior arm of the HSCC (5th position, Figure 1F); (7) the patient was slowly returned to the sitting position (Figure 1G).

The immediate treatment response was assessed by the supine head roll test 30 min after a single session of maneuver. The treatment was regarded as a success if the positional nystagmus was resolved or changed to a transient (lasts only a few seconds) geotropic form. If the first treatment was not successful, the patient received the same maneuver again. If the second trial was also unsuccessful, that was considered a treatment failure. Patients who did not respond to the treatment were scheduled for a follow-up visit one to four days later. At the first follow-up, the patients underwent a supine head roll test. The resolution of positional nystagmus or the change to transient geotropic nystagmus was considered a late resolution. The patients who still had apogeotropic nystagmus at the first follow-up again received the previously applied maneuver. In the same way, the maneuver was performed a maximum of twice, and the patients with treatment failure were scheduled for a second follow-up visit one to four days later. We collected the follow-up intervals only for patients who did not succeed immediately, and we could not collect the data from the patients who did not participate in the follow-up. The mean period between the first treatment and the first follow-up (the first interval) and between the first follow-up and the second follow-up (the second interval) in the total sample were 4.3 and 5.6 days, respectively. The first intervals in the CuRM group and the new maneuver groups were 6.2 days and 3.4 days (*p* = 0.036, Mann–Whitney U test), respectively, while the second intervals were 4.3 days and 7.5 days (*p* = 0.236, Mann–Whitney U test), respectively. All patients received no post-treatment instructions such as forced prolonged position or Brandt–Daroff exercises. We did not re-evaluate the patients who had a successful response to the treatment.

### 2.4. Statistical Analysis

Mann–Whitney U test was used to assess the differences in nonparametric data (age, mean symptom duration from onset to evaluation, and mean follow-up intervals) between the CuRM and new maneuver groups. Pearson’s chi-square test was used to assess the differences in categorical data (sex ratio, affected side ratio, and the immediate success rate) between the two groups. Fisher’s exact test was used to assess the difference in the late resolution rate at the first follow-up visit and the proportion of patients with a prior BPPV history between the two groups. All statistical analyses of the data were performed using SPSS version 22.0 (IBM SPSS Corp.; Armonk, NY, USA), and values of *p* less than 0.05 were considered significant.

## 3. Results

### 3.1. Patient Characteristics

Fifty-seven patients with HSCC cupulolithiasis (13 males and 44 females; age range, 13–77 years) were randomly allocated to the CuRM (*n* = 22) or the new maneuver (*n* = 35) for treatment. There were five (22.7%) males and seventeen (77.3%) females (mean age, 50.8 ± 14.6) in the CuRM group and eight (22.9%) males and twenty-seven (77.1%) females (mean age, 50.7 ± 11.9) in the new maneuver group. The affected side was on the right in 11 (50.0%) of 22 patients in the CuRM group, and 12 (34.3%) of 35 patients in the new maneuver group. The mean symptom duration of vertigo until the initial assessment was 2.43 ± 6.73 days in the CuRM group, and 2.86 ± 6.74 days in the new maneuver group. There was no significant difference in the mean age, sex ratio, affected side ratio, and duration of vertigo between the two groups. (Table 1) There were eight (14.0%) of 57 patients who had a history of BPPV prior to their first visit (two of twenty-two, 9.1%, in the CuRM group, and six of thirty-five, 17.1%, in the new maneuver group), and there were no significant differences in the proportion of patients with a prior BPPV history between the two groups (*p* = 0.466, Fisher’s Exact test).

### 3.2. Immediate and Short-Term Treatment Effects

In twenty-two patients who were treated with the CuRM, the immediate success rate was 36.4% (eight of twenty-two). Among the remaining 14 patients with treatment failure, two patients did not show up for their first follow-up visit, and 12 patients were assessed for a positional nystagmus test at the first follow-up. Apogeotropic nystagmus was not observed in 75% (nine of twelve) patients during the supine head roll test, of whom eight showed no nystagmus and one showed geotropic nystagmus. Three patients with apogeotropic nystagmus were treated with the CuRM at the first follow-up, which was not successful in any of the three patients. On the second follow-up visit, all three patients still showed apogeotropic nystagmus and were treated with the CuRM, which was not successful again in any of the three patients (Table 2 and Figure 2).

In thirty-five patients who were treated with the new maneuver, the immediate success rate was 28.6% (10 of 35). Among the remaining 25 patients with treatment failure, two patients did not show up for the first follow-up, and 23 patients were evaluated for a positional nystagmus test at the first follow-up. Apogeotropic nystagmus was not observed in 82.6% (19 of 23) patients during the supine head roll test, of whom 17 showed no nystagmus and two showed geotropic nystagmus. Four patients with apogeotropic nystagmus received the new maneuver at the first follow-up, which was successful in one patient. The remaining three patients with treatment failure were scheduled for a second follow-up. On the second follow-up visit, one patient did not show up, and two patients were assessed for a positional nystagmus test, and both showed no nystagmus (Table 2 and Figure 2).

There was no significant difference in the immediate success rate between the CuRM (eight of twenty-two, 36.4%) and the new maneuver (10 of 35, 28.6%) groups (*p* = 0.538, Pearson’s chi-square test). There was no significant difference in the late resolution rate at the first follow-up between the CuRM (9 of 12, 75%) and new maneuver (19 of 23, 82.6%) groups (*p* = 0.670, Fisher’s exact test).

## 4. Discussion

Although numerous maneuvers have been proposed to treat HSCC cupulolithiasis, the reported efficacies have been varied and a standard protocol has not been established yet. Most of the existing methods used inertia and gravity, or oscillation. However, the effect of vibration is a controversial topic. In the previous studies, mastoid oscillation was applied during the repositioning maneuver for BPPV, with the hypothesis that it might help detach the otoliths from the cupula or help the otoliths move into the utricle without adhering to the cupula or canal wall [9,20], but there are many reports that the application of vibration does not bring any therapeutic benefit [4,21,22,23,24]. In addition, it is reported that BPPV itself can occur after head and neck trauma or surgeries (such as nasal osteotomy or dental surgery) that cause impact or vibration [13,15,25,26,27].

Considering these controversies and the potential adverse effect of vibration on labyrinthine health, in this study, we designed a new maneuver conservatively using only inertia and gravity (by body repositioning and head rotation) to treat HSCC cupulolithiasis and compared its effect with the previously proposed maneuver using mastoid vibration. As our new maneuver does not use vibration, it can be performed in clinics that do not have oscillation equipment. Some early maneuvers for HSCC cupulolithiasis were limited in covering all possible locations of the otolithic debris in positional relation with the cupula (canal-side or utricular-side debris). For example, considering the current anatomical point of view, the Gufoni maneuver seems to be more effective for the utricular-side debris than the canal-side debris [1,16,28,29]. The Appiani maneuver, on the other hand, seems to be more effective for the canal-side debris than the utricular-side debris [1,30]. Our new maneuver for HSCC cupulolithiasis was designed focusing on the following considerations: (1) using only inertia and gravity and maximizing the effect, and (2) it should be effective for both the utricular-side and the canal-side debris. Currently, it is thought that the axis of the HSCC cupula is directed slightly lateral from the sagittal plane [19,20], and therefore a null point in the HSCC cupulolithiasis is usually observed with the head turned 20° to 30° to the affected side in the supine position [5]. Before rapid lying-down on the healthy or affected side, in the first and third position of our new maneuver, we tried to maximize the effect of inertia and gravity on the dislodged otoconia by turning the head to the affected side by 30° to ensure that the axis of the affected side cupula was parallel to the sagittal plane. In order to achieve the second consideration that it should be effective for the utricular-side and the canal-side debris, in our maneuver, we applied a rapid side-lying motion bidirectionally. These two considerations (30° head rotation to the affected side and thereafter bidirectional side-lying) are the main differences from the existing maneuvers such as the Gufoni maneuver, the Appiani maneuver, and Zuma’s maneuver. So, we think we can simply call our new maneuver a “bidirectional side-lying maneuver”.

In this study, the immediate success rates of CuRM and the new maneuver were 36.4% (8 of 22) and 28.6% (10 of 35), respectively (*p* = 0.538, Pearson’s chi-square test). This result appeared to be lower than that reported previously. However, when comparing the results, caution should be taken as each report differs in the way in which treatment success is defined. Kim et al. [9] reported that HSCC cupulolithiasis was resolved in 97.4% (76 of 78) of patients after an average of 2.1 repetitions of the maneuver, but this high remission rate was the result of up to a maximum of six treatments during a very long follow-up period (29.8 months, range 10–54 months). In Kim et al.’s report [9], the rate of remission after a single treatment was 61.5% (48 of 78). However, since they performed a head roll test at a follow-up two days after the first treatment and ended the treatment when no nystagmus or vertigo appeared, their successful cases might include patients with spontaneous remission during the follow-up. In addition to the definition of treatment success, diagnostic protocols, follow-up protocols, and additional instructions such as forced prolonged position or Brandt–Daroff exercises may affect the outcomes. Kim et al. [9] performed the head roll test three times during the diagnostic process, and after the maneuver, the patients were instructed to sleep in the lateral decubitus position on the healthy side. Repeated supine head roll tests may induce the detachment of otolith debris from the cupula [28,31], and a forced prolonged position itself may resolve the HSCC cupulolithiasis [32]. In this study, to confirm the effect of the maneuver itself, we performed the head roll test to a minimum during the diagnostic process, assessed the patients 30 min after the treatment, and did not give additional instructions to the patient after the treatment.

Another notable finding was that the late resolution rates at the first follow-up of the CuRM (75%, nine of twelve) and the new maneuver (82.6%, 19 of 23) were very high (Table 2). Several previous studies found that untreated HSCC cupulolithiasis had a short natural course. Shim et al. found that symptom remission took 3.7 days and the disappearance of positional nystagmus took 4.4 days [33], while Imai et al. found that symptom remission took 13 days in untreated HSCC cupulolithiasis [34]. It is thought that the head motions in daily life may induce the natural detachment of otoliths from the cupula, and the spontaneous dissolution of otoconia may also contribute to the short spontaneous remission [33]. In addition, it can be assumed that the time-dependent spontaneous dissolution of otoliths may affect the response to the repositioning maneuver, so the duration from the onset of BPPV to the treatment maneuver may affect the treatment response rate.

Our new treatment maneuver included fast side-lying on both the healthy side and the affected side. Before the side-lying, the head was turned 30° to the affected side to align the affected side cupula axis with the sagittal plane. Rapid deceleration and gravity help to detach the otolith debris from the cupula, specifically utricular-side debris when lying on the healthy side, and canal-side debris when lying on the affected side. Theoretically, a rapid acceleration in the initial stage of the side-lying may help detach the contralateral side otolith (canal-side debris when lying on the healthy side, and utricular-side debris when lying on the affected side), just as the rapid acceleration during upward head-turning helps to move the otolith to the opposite side of the acceleration. However, the effect of the acceleration that occurs during the rapid side-lying may not be very helpful, because it may be difficult to lay down as fast as the speed of turning the head. If our new repositioning maneuver does not work, it may be because the acceleration or deceleration was not fast enough. However, a recent study has shown that a faster execution of the Gufoni maneuver provides little benefit than a slower execution in treating apogeotropic HSCC BPPV [35], so this speculation remains debatable.

The location of the otolith in HSCC cupulolithiasis is a topic of interest. Kim et al. have noted the advantage of their repositioning maneuver, which provides an estimated location of the otolith [9]. Similarly, in our new maneuver, the immediate resolution of nystagmus after the maneuver may suggest utricular-side debris, and the transformation from apogeotropic nystagmus to geotropic form may suggest canal-side debris. However, the absence of response to the maneuver does not provide information on the location of the otolith. Therefore, we did not analyze the estimated location of the otoliths in this study.

The new maneuver we designed may have potential limitations. In order for the treatment maneuver to be widely used, it is advantageous for the maneuver to be simple. Although we have tried to keep the procedures as simple as possible, clinicians may need to become familiar with this new maneuver through repeated training before using it. Additionally, even if our new maneuver does not involve radical movements such as hopping, it may still be difficult for patients with limited physical activity to lie down or rotate their heads quickly. Lastly, in patients with HSCC cupulolithiasis, rapid side-lying on the healthy side may provoke more vertiginous symptoms than side-lying on the affected side only.

## 5. Conclusions

Our results showed that the new maneuver is an effective maneuver for treating HSCC cupulolithiasis with an immediate success rate of 28.6% (10 of 35). Although it did not show better results than the existing treatment method using vibration, there was no statistical difference in the immediate response rates between the two maneuvers. Considering the debate about the effectiveness of oscillation, we believe our new maneuver is another effective maneuver that uses only inertia and gravity, and it can be easily performed in clinics where oscillation equipment is not available.

## Figures and Tables

**Figure 1 jcm-11-04136-f001:**
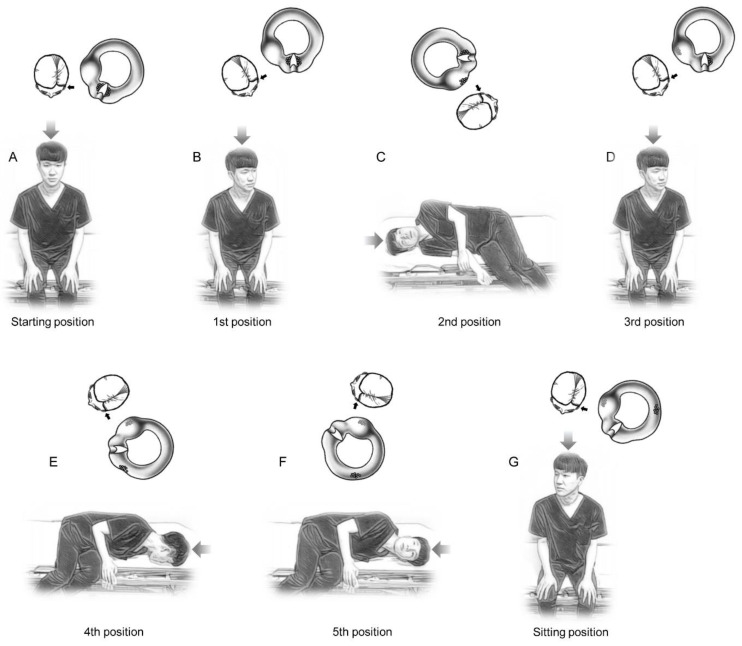
New repositioning maneuvers for treating left-sided horizontal semicircular canal (HSCC) cupulolithiasis. (**A**) The patient was seated in the center of the examination table. (**B**) The head was turned 30° to the affected side to align the affected side cupula axis to the sagittal plane (1st position). (**C**) The patient was rapidly brought down on the healthy side and held for two min in that position (2nd position). (**D**) The patient was slowly returned to the sitting position and kept in that position for 1 min (3rd position). (**E**) The head was turned 30° to the affected side, then the patient was quickly brought down on the affected side and held for two min (4th position). (**F**) The head was quickly turned 45° upward and kept in that position for 2 min (5th position). (**G**) The patient was slowly returned to the sitting position (Figure 1G). At the top of each posture, the left (small black arrow) HSCC (viewed from the top of the head, large gray arrow) are expressed, and the supposed movement and detachment of the otolithic debris from the cupula are also demonstrated.

**Figure 2 jcm-11-04136-f002:**
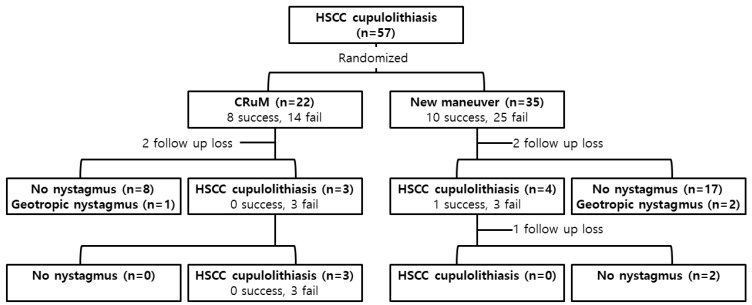
A summary of the treatment results through the follow-up. HSCC: horizontal semicircular canal, CuRM: cupulolith repositioning maneuver by mastoid oscillation.

**Table 1 jcm-11-04136-t001:** Subject characteristics (*n* = 57).

	CuRM (*n* = 22)	New Maneuver (*n* = 35)	*p*-Value
Age, mean ± SD	50.8 ± 14.6	50.7 ± 11.9	0.700 †
Sex, male:female	5:17	8:27	0.991 ‡
Affected side, right:left	11:11	12:23	0.239 ‡
Duration of vertigo, days	2.43 ± 6.73	2.86 ± 6.74	1.000 †

CuRM: cupulolith repositioning maneuver by mastoid oscillation. *p*-value < 0.05 was considered significant. † Mann–Whitney U-test. ‡ Pearson’s chi-square test.

**Table 2 jcm-11-04136-t002:** Treatment results of the CuRM and the new maneuver.

	CuRM (*n* = 22)	New Maneuver (*n* = 35)	*p* Value
Initial visit (immediate response)			
Success	8 (of 22, 36.4%)	10 (of 35, 28.6%)	0.538 †
Apogeotropic nystagmus	14 (of 22, 63.6%)	25 (of 35, 71.4%)	
First follow-up			
Follow-up loss	2	2	
Spontaneous resolution	9 (of 12, 75.0%)	19 (of 23, 82.6%)	0.670 ‡
Success	0 (of 3, 0%)	1 (of 4, 25.0%)	
Apogeotropic nystagmus	3 (of 3, 100%)	3 (of 4, 75.0%)	
Second follow-up			
Follow-up loss	0	1	
Spontaneous resolution	0 (of 3, 0%)	2 (of 2, 100%)	
Success	0 (of 3, 0%)	N/A	
Apogeotropic nystagmus	3 (of 3, 100%)	N/A	

CuRM: cupulolith repositioning maneuver by mastoid oscillation, N/A: not applicable. † Pearson’s chi-square test. ‡ Fisher’s exact test.

## Data Availability

Not applicable.

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
