# Peer review of "New Therapeutic Maneuver for Horizontal Semicircular Canal Cupulolithiasis: A Prospective Randomized Trial"

_jcm, 2022, doi:10.3390/jcm11144136_

Round 1
Reviewer 1 Report
Lee. et al. compare the efficacy of cupulolith repositioning manoeuvres in treating horizontal semicircular canal cupulolithiasis, namely the oscillation method, and a technique the authors have developed based on head and body motion/repositioning. Overall, the manuscript is relatively well written and displays data/results in a digestible format for readers. The strength of the work is the comparison of repositioning methods using a modest sample size (n=57), over almost a 2 year period. I believe the study provides utility and knowledge to the field and clinicians for the treatment of hSCC cupuloithiasis. Results support the use of a more conservative approach.
Despite this there are several limitations or comments which should be addressed prior to publication:
1. Despite being labelled as a ‘new technique’, it appears relatively similar to Zuma’s manoeuvre, involving sitting, quickly lying down and head rotations, etc. The authors should be clear to state how this technique is different to Zuma’s manoeuvre and other alike methods. Consider changing the name from ‘new maneuver’ to something more specific.
2. It is said throughout that the new technique relies on inertia and gravity. These are physical laws which govern the motion and position of bodies -rather the authors are using head and body rotation/movement/repositioning. Please explain more what you mean by this relative to altered labyrinthine health and disease, instead of just saying gravity.
3. Please provide references for some of the key background information within the introduction. E.g. lines 30-37.
4. If the video-Frenzel goggle technique has been used/published previously by the authors (or any other techniques) please reference.
5. Change ‘min’ to ‘minute’ or ‘minutes’ throughout the manuscript. E.g Line 118 – "help for two minutes".
6. Amend ‘wait’ to ‘waiting’ throughout the manuscript (Ctrl + F ‘wait’) – e.g. Line 122 "2 minutes of waiting in this step".
7. Line 148 – change ‘were’ to ‘was’.
8. Line 216 – change ‘did’ to ‘does’.
9. Provide references for key information in discussion section. Lines 222-225.
1. Consider adding ‘as rotation can be applied bidirectionally’ after “2) it should be effective for both the utricular side and the canal side debris”. Line 228.
1. Line 233 – “tried to maximise the effect of inertia and gravity” – please be more specific to state on what? The dislodged otoconia?
. Discussion: it would be welcome if potential limitations of the repositioning technique were discussed. I.e. the need for training of the maneuvers, quick-body motions on a dizzy patient, etc.

Author Response
[Response] We appreciate all of your valuable comments and suggestions. We have modified the manuscript according to the reviewer’s comments.

Reviewer 2 Report
The authors address an important issue for clinical vestibology concerning the treatment of BPPV from HSCC involvement. However, in some points they appear to be a bit approximate and a better description would facilitate the readers’ understanding.
ABSTRACT
Methods: Authors should provide a brief description of their new maneuver.
Conclusion: the sentences in the conclusion are redundant; correct English by synthesizing
INTRODUCTION
· A short sentence concerning the prevalence of the pathology in question (BPPV), its percentage of response compared to the more common Posterior SC BPPV is required to convey to readers the dimension of the problem
· Authors do not describe the difference between the geotropic form and the apogeotropic form of horizontal nystagmus. It would be right to clarify to the reader why the apogeotropic variant was chosen to experiment the new maneuver. A description to distinguish the healthy side from the affected side should also be provided in the introduction.
· Line 48: The authors list new treatment techniques for HSCC, but never mention the more well-known Semont maneuver, Asprella maneuver, Gufoni maneuver, Lempert maneuver-BBQ roll...please add
METHODS
· Line 105: “the head was turned 135 ÌŠ to the affected side” … is that number right? Explain what was the starting position. In CuRM you should describe first the movements of the body of the patient (from supine to lateral decubitus for example) and then the head position in a similar way to the “new maneuver” section, otherwise the passages from one position to another are not understandable.
· Description in the text should match those in the figure 1 (“the patient was slowly returned to the sitting position and kept in that position for 1 min (3rd position, Fig. 1-D)”…with the head turned 30° to the affected side? Please add. (“(G) the patient was slowly returned to the sitting position (Fig. 1-G)”)…with the head turned 30° to the healthy side? Please add.
· Line 145: “The treatment was regarded a success if the positional nystagmus was resolved or changed to transient geotropic form”, what do you mean with the term “transient”? Did you repeat the head roll test to observe the resolution of the geotropic nystagmus? Please describe better.
· Line 149 “The resolution of positional nystagmus or the change to transient geotropic nystagmus was considered a spontaneous resolution.” I disagree that the resolution can be defined “spontaneous” even if 4 days after the treatment. I would call it “late resolution” or something like that, but not spontaneous.
· Line 154: “The mean period between the first treatment and the first follow-up, and between the first follow-up and the second follow-up were 4.3 and 5.6 days” in the total sample? and in the two study groups? was it different?
STATISTICAL ANALYSIS
You didn’t mention the level of significance (p<0.05?)
RESULTS
· Please insert the age range and sex distribution in the total sample. Did you include children or only adult patients? otherwise enter in the exclusion criteria.
· Set the level of significance (x) in table 1 and replace the p-values ​​with ">x" when not significant
· Whenever the authors speak about significant or not significant results, they express improperly, because they didn’t express their level of significance. please add it in the methods and add p-values ​​in the results when you talk about comparisons.
· Line 205 “spontaneous resolution rate at the 1st follow-up”… here too I would substitute spontaneous with late resolution or something like that. I recommend that you also change it in the abstract.
Have you collected in the medical history if it was the first episode or a recurrence of BPPV? I think this is an important data to add because if there was a significant difference in terms of recurrence between the two groups, this could have influenced the results.

Author Response

(The authors gave the same response as above.)
